# Synergy by Ristocetin and CXCL12 in Human Platelet Activation: Divergent Regulation by Rho/Rho-Kinase and Rac

**DOI:** 10.3390/ijms24119716

**Published:** 2023-06-03

**Authors:** Yukiko Enomoto, Takashi Onuma, Takamitsu Hori, Kumiko Tanabe, Kyohei Ueda, Daisuke Mizutani, Tomoaki Doi, Rie Matsushima-Nishiwaki, Shinji Ogura, Hiroki Iida, Toru Iwama, Osamu Kozawa, Haruhiko Tokuda

**Affiliations:** 1Department of Neurosurgery, Gifu University Graduate School of Medicine, Gifu 501-1193, Aichi, Japan; 2Department of Anesthesiology and Pain Medicine, Gifu University Graduate School of Medicine, Gifu 501-1193, Aichi, Japan; 3Department of Pharmacology, Gifu University Graduate School of Medicine, Gifu 501-1193, Aichi, Japan; 4Department of Metabolic Research, Research Institute, National Center for Geriatrics and Gerontology, Obu 474-8511, Aichi, Japan; 5Department of Emergency and Disaster Medicine, Gifu University Graduate School of Medicine, Gifu 501-1193, Aichi, Japan; 6Department of Clinical Laboratory, National Center for Geriatrics and Gerontology, Obu 474-8511, Aichi, Japan

**Keywords:** CXCL12, ristocetin, Rho/Rho-kinase, Rac, soluble CD40 ligand, platelet

## Abstract

CXCL12, belonging to the CXC chemokine family, is a weak agonist of platelet aggregation. We previously reported that the combination of CXCL12 and collagen at low doses synergistically activates platelets via not CXCR7 but CXCR4, a specific receptor for CXCL12 on the plasma membrane. Recently, we reported that not Rho/Rho kinase, but Rac is involved in the platelet aggregation induced by this combination. Ristocetin is an activator of the von Willebrand factor that interacts with glycoprotein (GP) Ib/IX/V, which generates thromboxane A2 via phospholipase A2 activation, resulting in the release of the soluble CD40 ligand (sCD40L) from human platelets. In the present study, we investigated the effects of a combination of ristocetin and CXCL12 at low doses on human platelet activation and its underlying mechanisms. Simultaneous stimulation with ristocetin and CXCL12 at subthreshold doses synergistically induce platelet aggregation. A monoclonal antibody against not CXCR7 but CXCR4 suppressed platelet aggregation induced by the combination of ristocetin and CXCL12 at low doses. This combination induces a transient increase in the levels of both GTP-binding Rho and Rac, followed by an increase in phosphorylated cofilin. The ristocetin and CXCL12-induced platelet aggregation as well as the sCD40L release were remarkably enhanced by Y27362, an inhibitor of Rho-kinase, but reduced by NSC23766, an inhibitor of the Rac-guanine nucleotide exchange factor interaction. These results strongly suggest that the combination of ristocetin and CXCL12 at low doses synergistically induces human platelet activation via Rac and that this activation is negatively regulated by the simultaneous activation of Rho/Rho-kinase.

## 1. Introduction

Stroma cell-derived factor-I (CXCL12), a heparin-binding protein belonging to the CXC chemokine families, plays important roles in inducing and attracting immune cells, the homeostatic process, and pathological conditions [1]. Specific guanosine triphosphate (GTP)-binding-protein coupled receptors for CXCL12 such as CXCR4 and CXCR7 are functional receptors in several cells [2]. CXCR4 mainly regulates bone marrow-derived progenitors, circulating endothelial progenitor cells, smooth muscle progenitor cells, and platelets, whereas CXCR7chiefly regulates the development of the cardiac or central nervous system, the migration of primordial germ cells, and the promotion of tumor progression and metastasis [2]. Thus, CXCL12 is broadly involved in essential physiological processes such as embryogenesis, hematopoiesis, and angiogenesis, as well as pathological conditions such as cancer, neurodegenerative disorders, and atherosclerotic diseases [3,4]. Elevated plasma levels of CXCL12 are a biomarker for predicting acute coronary diseases and potential stroke [5,6,7]. In the hematopoietic system, megakaryocyte-lineage cells express CXCR4, and the CXCL12-CXCR4 axis is required for the migration of mature megakaryocytes into vascular niches, a crucial process in platelet biogenesis [8].

Platelets derived from the cytoplastic extension of mature megakaryocytes are essential for hemostasis and endothelium repair [9]. At sites of vessel wall injury, platelets in the blood flow are tethered to the subendothelial matrix via glycoprotein (GP) Ib/IX/V with the interaction of the von Willebrand factor (vWF), which elicits the secretion of autocrine/paracrine mediators, such as ADP and thromboxane A2 (TXA2) [9]. Subendothelial collagen exposed to the injured endothelium triggers the activation of platelets through GPVI and integrin α2β1 leading to platelet accumulation and adhesion, and GP IIb/IIIa is finally activated, followed by hemostatic plug formation [9,10]. In addition, platelets can be activated by shear stress under various pathophysiological conditions, such as atherosclerosis [11,12]. Under shear stress, platelet activation mainly depends on the interaction between vWF and GPIb/IX/V [11,12]. In contrast, human platelets store CXCL12 in the α-granule secretome and release it upon activation [2]. Platelet-derived CXCL12, which is a weak agonist of platelet activation, modulates the diverse functions of circulating platelets in an autocrine/paracrine manner via CXCR4 and CXCR7 expressed on the plasma membrane, such as chemotaxis, adhesion, proliferation, and the differentiation of nucleated cells [2]. This orchestrated process enhances the recruitment of platelets to injured vessel and tissue sites, promoting repair [2]. Regarding the cooperative effect of CXCL12 on platelet activation, we previously reported that CXCL12 with collagen at a low dose elicits platelet activation in the involvement of not CXCR7 but CXCR4 [13].

Rho and Rac, which belong to a superfamily of small molecular weight GTP-binding proteins called the Rho family, play important roles in several cellular functions, such as cytoskeletal reorganization and gene expression [14]. In the guanosine diphosphate (GDP)-binding forms, Rho and Rac are inactive, but are activated upon the exchange of GDP to GTP by the guanine nucleotide exchange factor [14]. We previously reported that Rho kinase, the downstream effector of Rho [14], and Rac are involved in platelet activation stimulated by TXA2 [15,16]. In addition, we recently reported that Rac is involved in the activation of human platelets stimulated by a combination of subthreshold doses of CXCL12 and collagen [17]. Ristocetin is known as an activator of vWF, that interacts with GPIb/IX/V, like a shear stress [18] and induces TXA2 synthesis via the activation of phospholipase A2 [12]. We previously reported that ristocetin-generated TXA2 can induce the release of soluble CD40 ligands from human platelets [19], in which Rho-kinase and Rac are involved [15,16]. Hence, we speculated that Rho/Rho-kinase and Rac play pivotal roles in the regulation of platelet function by ristocetin and CXCL12. However, the effects and mechanism of their combination on platelet function have not yet been elucidated.

In the present study, we investigated the effects of a combination of ristocetin and CXCL12 at low doses on human platelet activation and its underlying mechanisms. Our results strongly suggest that the combination synergistically induces human platelet activation via Rac, and that activation is negatively regulated by the simultaneous activation of Rho/Rho-kinase.

## 2. Results

### 2.1. Effect of Simultaneous Stimulation of Ristocetin and CXCL12 in Low Doses on Human Platelet Aggregation

First, we investigated the effect of ristocetin and CXCL12 combination at subthreshold levels on human platelet aggregation. To establish the subthreshold level, we examined the platelet aggregation stimulated by various doses (0.75 mg/mL, 0.80 mg/mL, 0.85 mg/mL, and 1.0 mg/mL) of ristocetin. Although high doses of ristocetin up to 0.85 mg/mL hardly increased the transmittance or size ratio of large aggregates, ristocetin at a dose of 1.0 ng/mL considerably induced platelet aggregation (Figure 1). Thereafter, we examined the effects of the simultaneous stimulation of various doses (3 ng/mL, 10 ng/mL, and 30 ng/mL) of CXCL12 with ristocetin at 0.85 mg/mL, the subthreshold level that is insufficient to induce platelet aggregation by itself, on the platelet aggregation. CXCL12 at a dose of up to 3 ng/mL in combination with ristocetin hardly increased the transmittance or size ratio of large aggregates; however, CXCL12 at 10 or 30 ng/mL in combination with ristocetin induced platelet aggregation in a dose-dependent manner (Figure 2A). We further examined the effect of the simultaneous stimulation of ristocetin (0.75 mg/mL, 0.80 mg/mL, 0.85 mg/mL, and 1.0 mg/mL) with CXCL12 at 30 ng/mL on the platelet aggregation. Although CXCL12 alone induced an increase in small aggregates, CXCL12 significantly enhanced platelet aggregation stimulated by subthreshold levels of ristocetin at a dose between 0.75 mg/mL and 0. 85 mg/mL (Figure 2B).

### 2.2. Effect of Anti-CXCR4 or Anti-CXCR7 Monoclonal Antibody on the Human Platelet Aggregation Induced by Simultaneous Stimulation of Ristocetin and CXCL12 

Human platelets express two specific receptors for CXCL12, namely CXCR4 and CXCR7 [2]. To clarify which receptor(s) are involved in the synergistic effect of CXCL12 and ristocetin on platelet aggregation, we examined whether an antibody against CXCR4/CXCR7 affects aggregation stimulated by a combination of CXCL12 and ristocetin at subthreshold levels. The antibody against CXCR4 markedly suppressed platelet aggregation induced by the combination treatment, but the antibody against CXCR7 had no effect (Figure 3).

### 2.3. Effect of Simultaneous Stimulation of Ristocetin and CXCL12 on Rho/Rho-Kinase and Rac Activation in Human Platelets

We have previously reported that Rho/Rho-kinase and Rac are involved in platelet activation stimulated by TXA2 [15,16], resulting from GP Ib/IX/V activation by ristocetin [20]. Recently, we showed that Rac (and not Rho/Rho kinase) plays a role in the activation of human platelets stimulated by a combination of a subthreshold dose of CXCL12 and collagen [17]. Based on these findings, we examined whether simultaneous stimulation with ristocetin and CXCL12 at low doses induced the activation of Rho/Rho-kinase and/or Rac in human platelets.

Although ristocetin (1.0 mg/mL) or CXCL12 (30 ng/mL) alone did not increase the levels of binding Rho, the combination remarkably increased the levels of binding Rho 1 min after stimulation (Figure 4). Similarly, the single administration of ristocetin (0.90 mg/mL) or CXCL12 (30 ng/mL) did not increase the levels of GTP-binding Rac1, but the simultaneous administration of markedly increased GTP-binding Rac1 at 1 min after the stimulation (Figure 5).

### 2.4. Effect of Y27632 on Platelet Aggregation Induced by Simultaneous Stimulation of Ristocetin and CXCL12

To determine whether Rho activation is involved in platelet activation induced by the combination of ristocetin and CXCL12 at subthreshold levels, we investigated the effect of Y27632, a Rho-kinase inhibitor [21], on platelet aggregation. Y27632 (30 μM) did not suppress but caused rather an increase in transmittance with an increased ratio of large aggregates and decreased ratio of small aggregates stimulated by the combination (Figure 6A). We confirmed that Y27632 reduced the phosphorylation of cofilin, a substrate of Rho-kinase [22] upregulated in platelets stimulated by the combination of ristocetin and CXCL12 at low doses (Figure 6B).

### 2.5. Effect of NSC23766 on Platelet Aggregation Induced by Simultaneous Stimulation of Ristocetin and CXCL12

To determine whether Rac activation is involved in the activation of platelets induced by the combination of ristocetin and CXCL12 at subthreshold levels, we investigated the effect of NSC23766, a selective inhibitor of guanine nucleotide exchange factor interactions [23], on platelet aggregation induced by the combination. Although NSC23766 at 1 μM hardly affected platelet aggregation, NSC23766 at 3 μM dose-dependently inhibited platelet aggregation, and the effect of NSC23766 was almost complete at 7 μM (Figure 7A). We also confirmed that NSC23766 reduced the levels of GTP-binding Rac, which was upregulated in platelets stimulated with the combination of ristocetin and CXCL12 at low doses (Figure 7B).

### 2.6. Effects of Y27632 or NSC23766 on the Release of sCD40 Ligand from Platelets Induced by the Simultaneous Stimulation of Ristocetin and CXCL12

We further examined the effects of Y27632 and NSC23766 on the release of sCD40 ligand from platelets activated by simultaneous stimulation with ristocetin and CXCL12. Although the single administration of ristocetin (0.70–1.0 mg/mL, a dose presented 70% transmittance on platelet aggregation) or CXCL12 (30 ng/mL) did not increase the release, the simultaneous administration of them, as well as platelet aggregation, significantly increased the sCD40 ligand release from platelets (Figure 8A,B). On the other hand, Y27632 (30 μM) significantly upregulated the sCD40 ligand release from platelets activated by the simultaneous administration of ristocetin and CXCL12 (30 μM) (Figure 8A). However, NSC23766 (7 μM) remarkably suppressed the sCD40 ligand release from the platelets activated by the combination of ristocetin and CXCL12 in the low doses (Figure 8B).

## 3. Discussion

In the present study, we investigated the effects and mechanism of simultaneous stimulation of human platelets with subthreshold concentrations of ristocetin and CXCL12. Using the aggregometer based on a laser scattering method, we found that ristocetin (0.85 mg/dL), which by itself was not able to elicit platelet aggregation, clearly induced the aggregation with an increase of large aggregates in the combination of simultaneous stimulation with CXCL12 (30 ng/mL); this alone enabled us to increase the microaggregates just a little. Considering that ristocetin is recognized as an activator of GPIb/IX/V that interacts with vWF [18], it is likely that the GPIb/IX/V activation of platelet aggregation could be enhanced by synergistic stimulation with CXCL12. In addition, we found that the antibody against not CXCR7 but CXCR4 suppressed platelet aggregation induced by the combination of ristocetin and CXCL12 at threshold levels, suggesting the specific involvement of CXCR4 in the amplifying effect of CXCL12. Therefore, it is most likely that the synergistic effects of ristosetin and CXCL12 on platelet activation are mediated by GPIb/XI/V and CXCR4. We recently reported that subthreshold concentrations of collagen and CXCL12 show synergistic effects on platelet aggregation mediated by GPVI and CXCR4, respectively, but that the combination of ADP or thrombin with CXCL12 does not amplify it [13]. Thus, CXCL12 could synergistically enhance the platelet response to GPVI and GPIb/IX/V activation, and this effect was mainly mediated by CXCR4.

Based on our previous findings that both Rho/Rho-kinase and Rac are involved in ristocetin-induced platelet activation [15,16], we further investigated the mechanism underlying the synergistic effect of ristocetin and CXCL12 at subthreshold levels. We found that the levels of both Rho and Rac in their GTP-binding forms were considerably increased in platelets stimulated by the combination of ristocetin and CXCL12 at low concentrations, which by themselves had no effect. Thus, we examined the effect of Y27632 [21] and found that an increase in transmittance was associated with an increase in the large aggregate ratio and that a decrease in the small aggregate ratio was observed in Y27632-treated platelets stimulated by the combination of ristocetin and CXCL12 at low doses, that is, the inhibitor of Rho could upregulate platelet aggregation. Because we confirmed the suppression of cofilin phosphorylation by Y27632, its effect on platelet aggregation was thought to be caused by the suppression of Rho/Rho-kinase. Therefore, it is likely that Rho/Rho-kinase—which is activated by the combination of ristocetin and CXCL12 at subthreshold levels—plays a role as it negatively regulates platelet activation. Next, we examined the effect of NSC23766 [23] and found that the platelet aggregation stimulated by this combination was considerably suppressed by NSC23766. Rac, in contrast to Rho, positively regulates the platelet activation stimulated by the combination. We further examined the effects of Y27632 and NSC23766 on the release levels of the sCD40 ligand from platelets simultaneously stimulated by the combination and found that the release was significantly upregulated by Y27632 but suppressed by NSC23766. It is most likely that Rac acts as a positive regulator, whereas Rho acts as a negative regulator of platelet activation stimulated by the combination. Regarding the roles of Rho and Rac, we recently reported that Rac is involved in platelet activation induced by a combination of collagen and CXCL12 at subthreshold levels [17], which is quite different from the present findings in terms of Rho. To the best of our knowledge, this is the first report to clearly indicate the role of Rho and Rac as breaks and accelerators in the regulation of human platelets. The regulatory mechanisms are summarized in Figure 9.

From the perspective of pathological relevance, CXCL12 is highly expressed in smooth muscle cells, endothelial cells, and macrophages in atherosclerotic plaques [24], where the shear stress is amplified by an irregular bloodstream. Thus, platelets can be easily activated by CXCL12 and shear stress-induced GP Ib/IX/V activation via the mechanism shown here, leading to aggregation. Moreover, the activated platelets can release CXCL12 by themselves [2], resulting in an augmentation in the synergistic effect with GPIb/IX/V, which may cause thrombus formation in acute occlusive angiopathies, such as brain infarction and acute coronary syndrome. Elevated plasma levels of CXCL12 are biomarkers for predicting acute coronary disease and future stroke [5,6,7]. The synergistic effect of the shear stress, which was mimicked by ristocetin, and CXCL12 on platelet activation appears to be the mechanism underlying vascular diseases. In addition, fasudil, the inhibitor of Rho-kinase [21], is currently established as a useful therapeutic tool for cerebral vasospasm after subarachnoid hemorrhage [25]. Considering our present findings that Rho/ROCK can negatively regulate platelet activation, neurosurgeons should pay attention to platelet function when using this agent. A limitation of our study is that it was based on ex vivo experiments. It is worth considering whether a proteomic study analyzing clots in experimental groups could provide additional insights and a clearer understanding of the downstream pathways and proteins involved in platelet activation. Further investigations are required to clarify the relevance of platelet activation by CXCL12 and shear stress as well as the function of Rac and Rho-kinase in platelets.

## 4. Material and Methods

### 4.1. Materials

Ristocetin was purchased from Sigma-Aldrich (Darmstadt, Germany). Recombinant CXCL12, mouse anti-CXCR4 monoclonal antibody, mouse anti-CXCR7 monoclonal antibody, and the soluble CD40 ligand (sCD40L) ELISA kit were purchased from R & D Systems, Inc. (Minneapolis, MN, USA). NSC23766 and Y27632 were purchased from Tocris Bioscience (Bristol, UK) and Calbiochem/Novabiochem Co. (La Jolla, CA, USA), respectively. GAPDH rabbit polyclonal antibody (cat. no. SC-25778) was purchased from Santa Cruz Biotechnology, Inc. (Dallas, TX, USA). Phospho-specific cofilin antibodies were obtained from Cell Signaling Technology, Inc. (Danvers, MA, USA). Rac1 and Rho activation assay kits were purchased from Millipore (Billerica, MA, USA). Other materials and chemicals were obtained from commercial sources. Ristocetin was dissolved in dimethyl sulfoxide. The maximum concentration of dimethyl sulfoxide was 0.1%, which did not affect the platelet aggregation.

Y27632 and NSC23766 were dissolved in dimethyl sulfoxide. The maximum concentration of dimethyl sulfoxide was 0.3%, which did not affect the platelet aggregation, protein detection by Western blotting or ELISA for sCD40L.

### 4.2. Preparation of Platelets

Whole blood samples were obtained from healthy volunteers. They were immediately added to 1/10 volume of 3.8% sodium citrate after blood collection. Platelet-rich plasma (PRP) was obtained by centrifugation at 155× *g* for 12 min at room temperature. The residual samples were centrifuged again at 1400× *g* for 5 min to obtain (PPP). This study was approved by the Committee of Human Research at Gifu University Graduate School of Medicine (approval code: 2020-133, approval date: 2 September 2020). Written informed consent was obtained from all the participants.

### 4.3. Platelet Aggregation

Platelet aggregation was measured using citrated PRP in a light transmittance aggregometer (PA-200: Kowa Co. Ltd., Tokyo, Japan) with a laser-scattering system which can determine the particle size of platelet aggregation (small; 9–25 μm, medium; 25–50 μm and large; 50–70 μm). The percentage transmittance of non-stimulated PRP was recorded as 0%, and that of the corresponding PPP (blank) was recorded as 100%. PRP was pre-incubated at 37 °C for 1 min at a stirring speed of 800 rpm and stimulated by CXCL12 with ristocetin. When indicated, PRP was pretreated with anti-CXCR4 monoclonal antibody, anti-CXCR7 monoclonal antibody, control IgG, or Y27632 for 15 min and then stimulated with CXCL12 and/or ristocetin. Platelet aggregation was monitored for 4 min. The reaction was terminated by adding ice-cold EDTA (10 mM) [17].

### 4.4. Protein Preparation after Stimulation

After the termination of platelet aggregation by the addition of ice-cold EDTA (10 mM), the mixture was centrifuged at 10,000× *g* at 4 °C for 2 min. The supernatant was stored at −80 °C for ELISA analysis. For Rho or Rac activity assays, the pellet was washed twice with ice-cold Tris-buffered saline (TBS) [17]. For Western blot analysis, the pellet was washed twice with ice-cold phosphate-buffered saline (PBS) [17].

### 4.5. Analysis of Rho or Rac Activity

The TBS-washed pellet was lysed by sonication in a Mg^2+^ lysis buffer (MLB). GTP-binding Rho or Rac was immunoprecipitated using a Rho or Rac1 Activation Assay Kit according to the manufacturer’s instructions (Millipore, Billerica, MA, USA). The immunoprecipitated GTP-binding Rho or Rac and the immunoprecipitated lysates, namely total Rho or Rac, were subjected to Western blot analysis using antibodies against Rho or Rac1 [17].

### 4.6. Western Blotting

Western blot analysis was performed as described previously [17]. Briefly, the PBS-washed pellet was lysed by boiling it in lysis buffer [62.5 mM Tris-HCl, pH 6.8, 2% sodium dodecyl sulfate (SDS), 50 mM dithiothreitol, and 10% glycerol]. Lysate-applied SDS-polyacrylamide gel electrophoresis (PAGE) was performed as previously described by Laemmli on 10% or 12.5% polyacrylamide gel [26]. The proteins in the gel were transferred onto a PVDF membrane and blocked with 5% fat-free dry milk in PBS containing 0.1% Tween 20 (PBS-T), 10 mM Na_2_HPO_4_, 1.8 mM KH_2_PO_4_ (pH 7.4), 137 mM NaCl, 2.7 mM KCl, and 0.1% Tween 20 for 2 h. After incubation with the indicated primary antibodies, peroxidase-labeled anti-rabbit IgG antibodies were used as the secondary antibodies. Both the primary and secondary antibodies were diluted to optimal concentrations with 5% fat-free dry milk in PBS-T. Peroxidase activity on the PVDF membrane was imaged on an X-ray film using an ECL Western blotting detection system (GE Healthcare, Buckinghamshire, UK) as described in the manufacturer’s protocol. Densitometric analysis was performed using ImageJ software (version 1.50; NIH, Bethesda, MD, USA). The levels of phosphorylation were evaluated by the background-subtracted intensity of each signal, which was normalized to the respective intensity of GAPDH and plotted as a fold increase compared to the control [17].

### 4.7. ELISA for sCD40L

The levels of sCD40L in the supernatant of the conditioned mixture after platelet aggregation analysis were determined using an ELISA kit for sCD40L in accordance with the manufacturer’s instructions [19].

### 4.8. Statistical Analysis

The data are presented as mean ± SEM. Data were analyzed using the Mann–Whitney U test with JMP v13.0.0 (SAS Institute, Inc., Cary, NC, USA), and *p*-value < 0.05 were considered statistically significant.

## 5. Conclusions

In conclusion, our results strongly suggest that ristocetin and CXCL12 synergistically induce platelet activation via Rac, which is negatively regulated by the simultaneous activation of Rho/Rho-kinase. The findings of the present study provide important insights into the mechanisms underlying thrombus formation in atherosclerotic diseases involving shear stress.

## Figures and Tables

**Figure 1 ijms-24-09716-f001:**
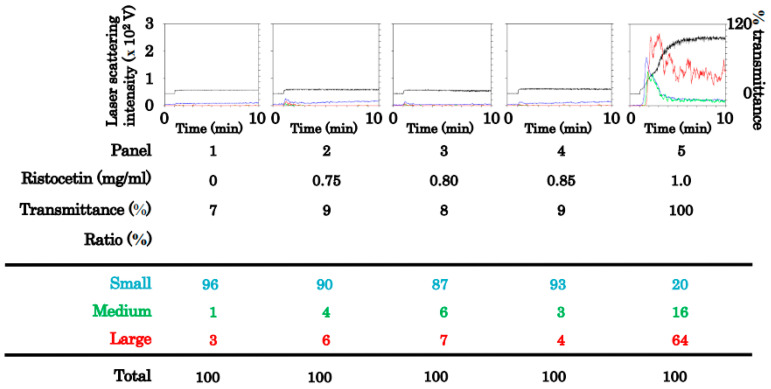
Platelet aggregation induced by various doses of ristocetin for determination of the subthreshold dose. Platelet-rich plasma was simulated with 0.75 mg/mL, 0.80 mg/mL, 0.85 mg/mL, and 1.0 mg/mL of ristocetin for 5 min. Black line indicates the percentage of transmittance of each sample (isolated platelets recorded as 0%, and platelet-poor plasma as 100%). Blue line indicates small aggregates (9–25 μm). Green line represents medium aggregates (25–50 μm), and red line represents large aggregates (50–70 μm). The lower panel presents the distribution (%) of aggregated particle size as measured by laser-scattering.

**Figure 2 ijms-24-09716-f002:**
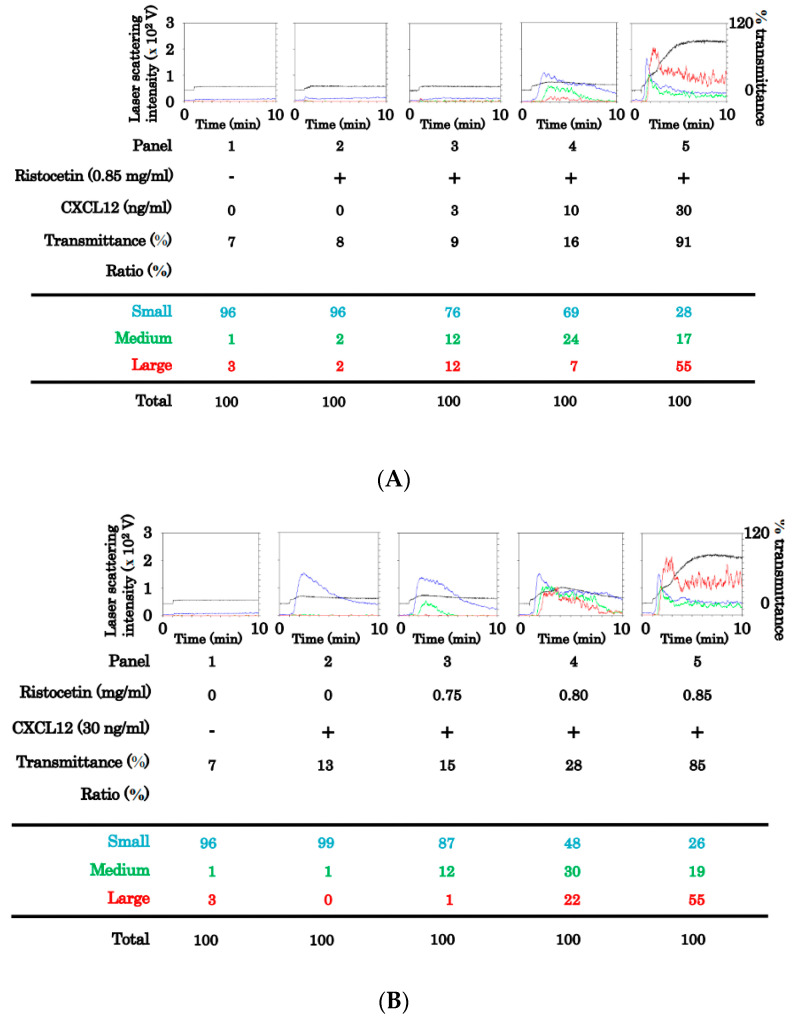
Platelet aggregation induced by simultaneous stimulation of ristocetin and CXCL12 in the low doses. (**A**) Platelet-rich plasma (PRP) was simultaneously stimulated with different doses (3 ng/mL, 10 ng/mL, and 30 ng/mL) of CXCL12 with the subthreshold dose (0.85 mg/mL) of ristocetin for 5 min. (**B**) PRP was simultaneously stimulated by different doses (0.75 mg/mL, 0.80 mg/mL, and 0.85 mg/mL) of ristocetin with a fixed dose (30 ng/mL) of CXCL12. Black line indicates the percentage of transmittance of each sample (isolated platelets recorded as 0%, and platelet-poor plasma as 100%). Blue line indicates small aggregates (9–25 μm), green line represents medium aggregates (25–50 μm), and red line represents large aggregates (50–70 μm). The lower panel presents the distribution (%) of aggregated particle size as measured using laser-scattering method.

**Figure 3 ijms-24-09716-f003:**
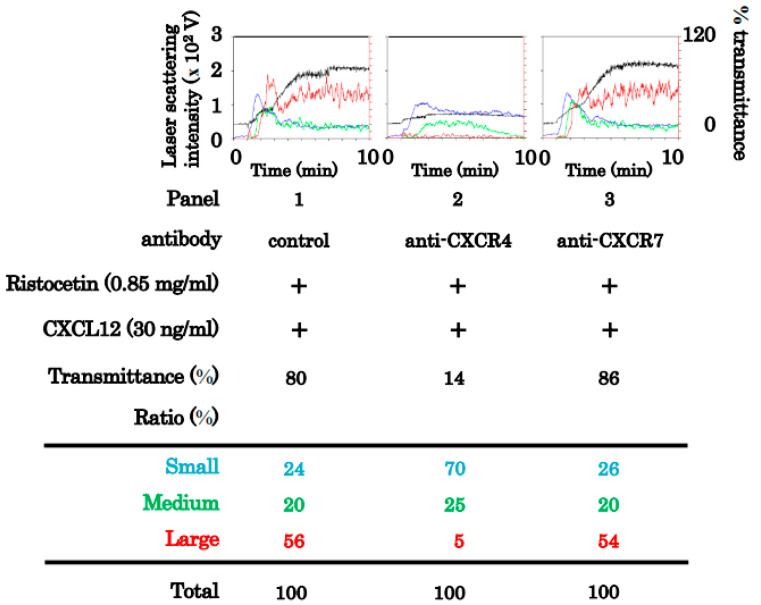
Effects of anti-CXCR4 or anti-CXCR7 monoclonal antibody on the platelet aggregation induced by the simultaneous stimulation of ristocetin and CXCL12 in low doses. Platelet-rich plasma was pretreated with 10 μg/mL of control IgG, anti-CXCR4 monoclonal antibody or anti-CXCR7 monoclonal antibody for 3 min, and then simultaneously stimulated by 30 ng/mL of CXCL12 and 0.85 mg /mL of ristocetin for 5 min. Black line indicates the percentage of transmittance of each sample (isolated platelets recorded as 0%, and platelet-poor plasma as 100%). Blue, green, and red lines represent small aggregates (9–25 μm), medium aggregates (25–50 μm), and large aggregates (50–70 μm), respectively. The lower panel presents the distribution (%) of aggregated particle size as measures using laser-scattering method.

**Figure 4 ijms-24-09716-f004:**
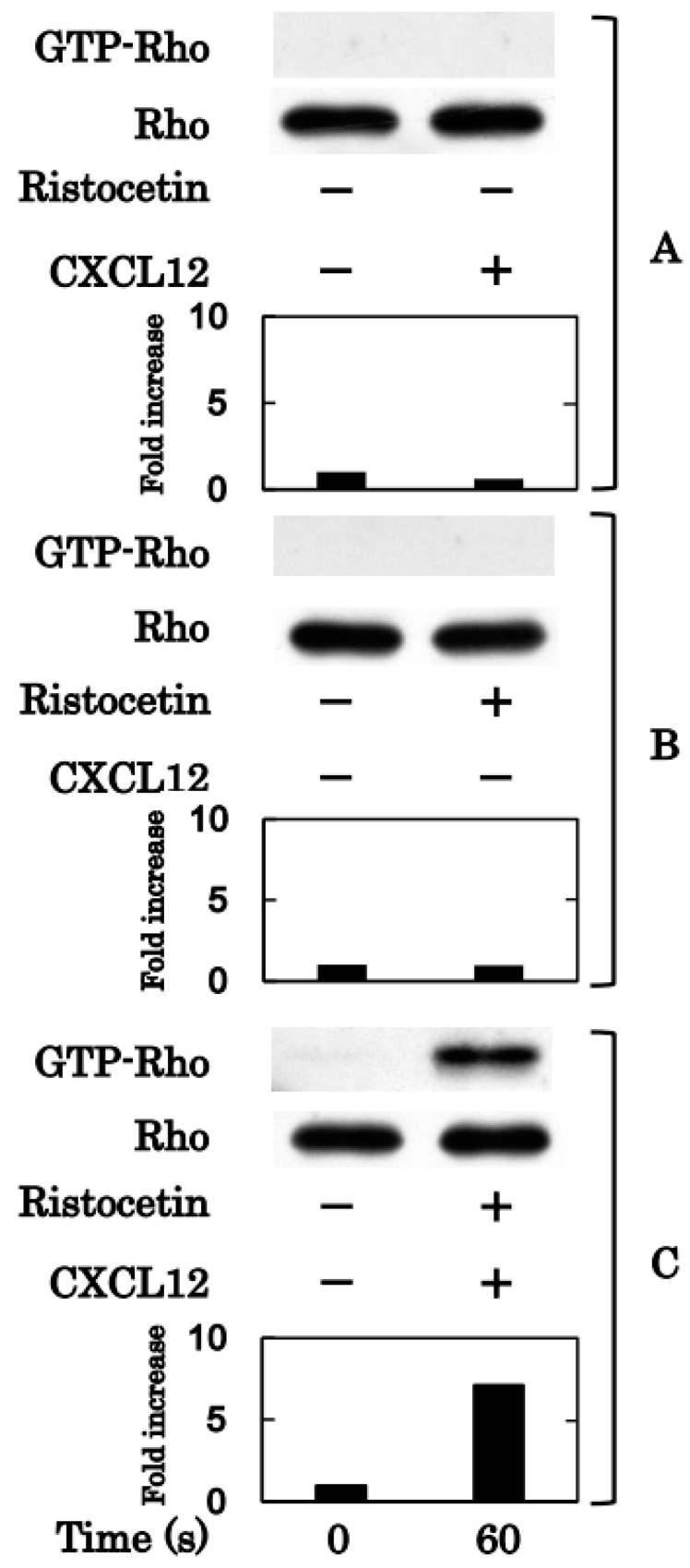
Effects of simultaneous stimulation of ristocetin and CXCL12 on Rho/Rho-kinase in human platelets. Platelet-rich plasma was stimulated by 30 ng/mL of CXCL12 (**A**), 1.0 mg/mL of ristocetin (**B**) or their combination (**C**) for 60 s. The reaction was terminated by adding ice-cold EDTA (10 mM) solution. Guanosine triphosphate (GTP)-Rho was immunoprecipitated using the Rho activation assay kit. The immunoprecipitated GTP-Rho and pre-immunoprecipitated lysate (Rho) were subjected to sodium dodecyl sulfate-polyacrylamide gel electrophoresis (SDS-PAGE) using antibodies against Rho. The histogram shows the fold increase from unstimulated cells; data were obtained with laser densitometric analysis.

**Figure 5 ijms-24-09716-f005:**
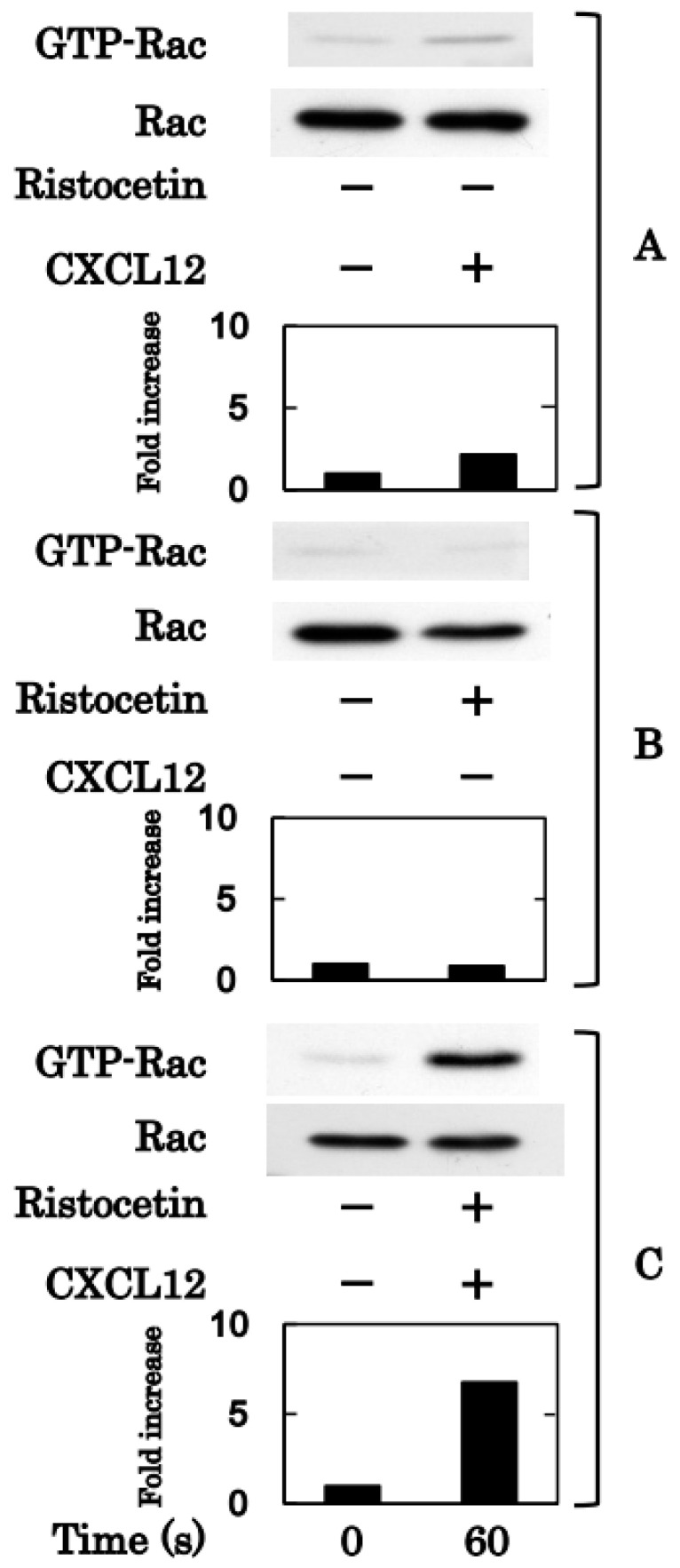
Effects of simultaneous stimulation of ristocetin and CXCL12 on Rac in human platelets. PRP was stimulated by 30 ng/mL of CXCL12 (**A**), 0.9 mg/mL of ristocetin (**B**) or their combination (**C**) for 60 s. The reaction was terminated by adding ice-cold EDTA (10 mM) solution. Guanosine triphosphate (GTP)-Rac was immunoprecipitated using the Rac1 activation assay kit. The immunoprecipitated GTP-Rac and pre-immunoprecipitated lysate (Rac) were subjected to sodium dodecyl sulfate-polyacrylamide gel electrophoresis (SDS-PAGE) using antibodies against Rac. The histogram shows the fold increase from unstimulated cells; data were obtained using laser densitometric analysis.

**Figure 6 ijms-24-09716-f006:**
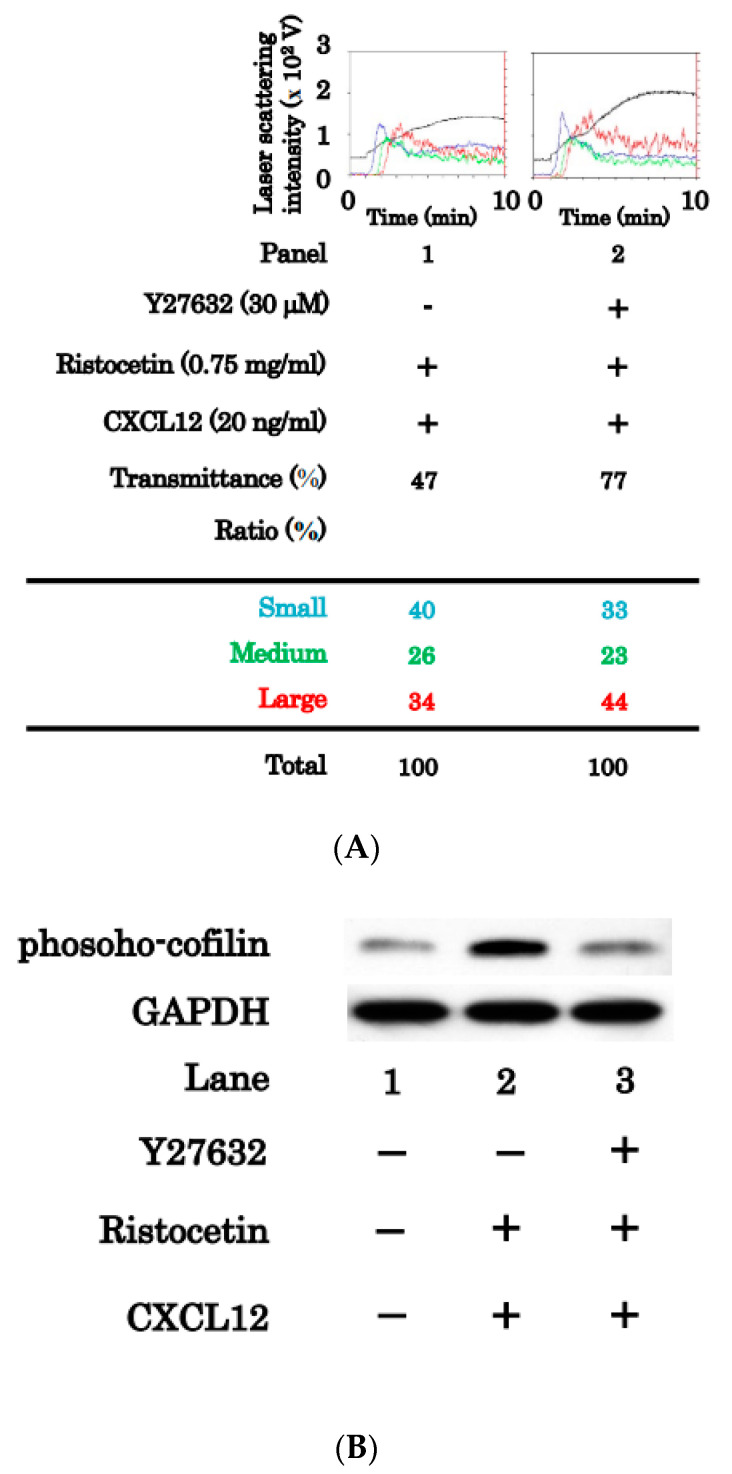
Effects of Y27632 on the platelet aggregation and the phosphorylation of cofilin induced by the simultaneous stimulation of ristocetin and CXCL12 in low doses. Platelet-rich plasma was pretreated with 30 μM of Y27632 or vehicle for 15 min, and then simultaneously stimulated by 0.75 mg/mL of ristocetin and 20 ng/mL CXCL12 for 10 min (**A**) or 1 min (**B**). The reaction was terminated by the addition of ice-cold EDTA (10 mM) solution. (**A**) Black line indicates the percentage of transmittance of each sample (isolated platelets recorded as 0%, and platelet-poor plasma as 100%). Blue, green, and red lines indicate small aggregates (9–25 μm); medium aggregates (25–50 μm); and larger aggregates (50–70 μm), respectively. The lower panel presents the distribution (%) of aggregated particle size as measures by laser-scattering. (**B**) The lysed platelets were subjected to Western blot analysis using antibodies against phospho-specific cofilin, total cofilin or GAPDH.

**Figure 7 ijms-24-09716-f007:**
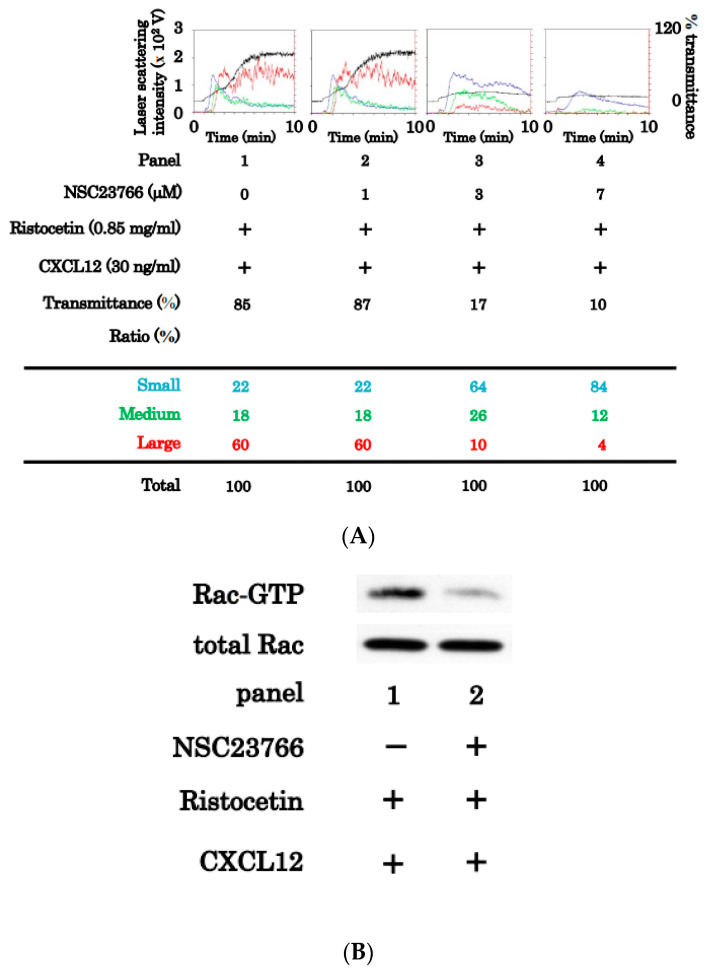
Effects of NSC23766 on the platelet aggregation and the level of Guanosine triphosphate (GTP)-Rac induced by simultaneous stimulation of ristocetin and CXCL12 in low doses. Platelet-rich plasma was pretreated with 1 μM, 3 μM or 7 μM of Y23766 or vehicle for 15 min, and then simultaneously stimulated by 0.85 mg/mL of ristocetin and 30 ng/mL CXCL12 for 10 min (**A**) or 1 min (**B**). The reaction was terminated by the addition of ice-cold EDTA (10 mM) solution. (**A**) Black line indicated the percentage of transmittance of each sample (isolated platelets recorded as 0%, and platelet-poor plasma as 100%). Blue, green, and red lines indicate small aggregates (9–25 μm); medium aggregates (25–50 μm); and larger aggregates (50–70 μm), respectively. The lower panel presents the distribution (%) of aggregated particle size as measured by laser-scattering. (**B**) Guanosine triphosphate (GTP)-Rac was immunoprecipitated using the Rac1 Activation assay kit. The immunoprecipitated GTP-Rac and pre- immunoprecipitated lysate (Rac) were subjects to SDS-PAGE using antibodies against Rac.

**Figure 8 ijms-24-09716-f008:**
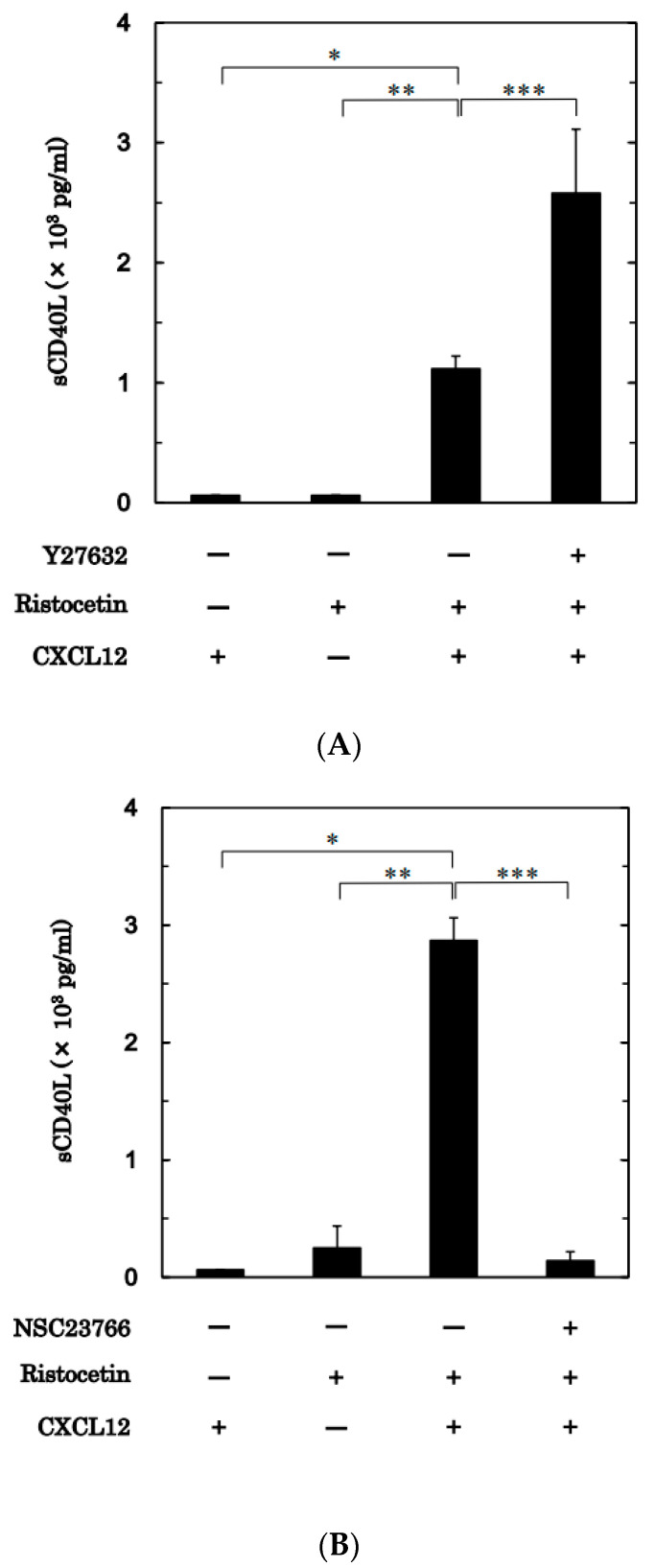
Effects of Y27632 and NSC23766 on the released levels of sCD40 ligand from platelets induced by the simultaneous stimulation of ristocetin and CXCL12 in low doses. Platelet-rich plasma was pretreated with 30 μM of Y27632 (**A**), 7 μM of NSC23766 (**B**) or vehicle for 15 min, and then stimulated by ristocetin alone or the combination of CXCL12 (30 ng/mL) for 15 min. The dose of ristocetin was adjusted individually to achieve a percentage transmittance of 70–100% recorded by the aggregometer (0.70–1.0 mg/mL) in combination with 30 ng/mL of CXCL12. The reaction was terminated by the addition of ice-cold EDTA (10 mM) solution. The conditioned mixture was centrifuged at 10,000× *g* at 4 °C for 2 min, and the supernatant was then subjected to an enzyme-linked immunosorbent assay for sCD40L. * *p* < 0.05, ** *p* < 0.01, *** *p* < 0.001.

**Figure 9 ijms-24-09716-f009:**
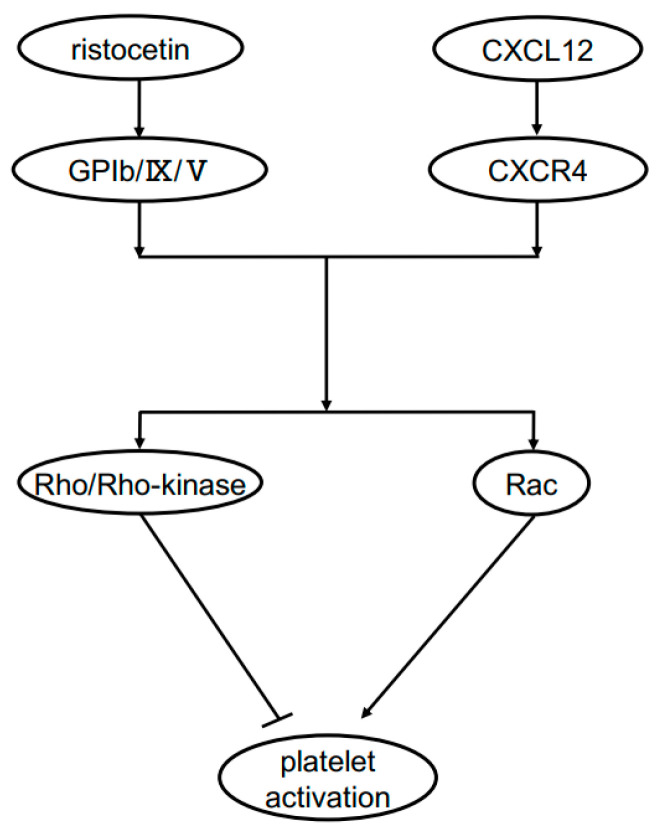
Schematic representation of the mechanism of synergistic effect of ristocetin and CXCL12 on platelet activation.

## Data Availability

The datasets generated and/or analyzed in this study are available from the corresponding author on reasonable request.

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
