# Peer review of "Synergy by Ristocetin and CXCL12 in Human Platelet Activation: Divergent Regulation by Rho/Rho-Kinase and Rac"

_ijms, 2023, doi:10.3390/ijms24119716_

Round 1

Reviewer 1 Report

In this manuscript the authors show that CXCL12 synergizes with ristocetin to induce platelet aggregation, which is positively regulated by Rac but negatively regulated by Rho/Rho-kinase.

A main weakness of the presentation is that the rationale for this study is not clearly explained in the Introduction section. In this regard, references 23-25 should be mentioned in the Introduction section. Ristocetin per se is not physiologically relevant. Its study is of interest solely because it is generally believed that ristocetin simulates the effect of shear stress on von Willebrand factor.

Instead, the authors state “Ristocetin is an activator of glycoprotein (GP) lb/IX/V to interact with von Willebrand factor,...”. (Lines 20-21) and “Ristocetin is known as an activator of GP lb/IX/V to interact with vWF…..,”. (Line 81) These statements are not supported by the cited reference (#15) and are not consistent with the prevailing concept that ristocetin is an activator of von Willebrand factor. In ristocetin cofactor activity assay, a laboratory test to measure the activity of von Willebrand factor in plasma, ristocetin is added to formalin fixed platelets suspected in test plasma to induce platelet aggregation. In this process, GPIb/IX/V is cannot be activated since the platelets are formalin fixed. If the authors are aware of any credible literature that supports their claim, it should be cited.

The entire manuscript should be proof-read by someone with fluency in English. As an example, "Ristocetin is an activator of glycoprotein (GP)lb/IX/V to interact with von Willebrand factor, which generates thromboxane A2....." (Lines 20-21)

Author Response

Dear Reviewer #1

Thank you for your fruitful comments.  We revised the manuscript according to your comments as follows:

Comment 1) In this manuscript the authors show that CXCL12 synergizes with ristocetin to induce platelet aggregation, which is positively regulated by Rac but negatively regulated by Rho/Rho-kinase.A main weakness of the presentation is that the rationale for this study is not clearly explained in the Introduction section. In this regard, references 23-25 should be mentioned in the Introduction section.

Our response:

As pointed out, we moved the sentence from Discussion to Introduction in the revised manuscript as follows:

*Line 47-48; "Elevated plasma levels of CXCL12 is a biomarker for predicting acute coronary diseases and potential stroke [5-8]."

According to the revision, Ref. 23, Ref. 24 and Ref. 25 in the original manuscript were renumbered to Ref. 5, Ref. 6, and Ref. 7 in the revised manuscript. 

In addition, subsequent references have been renumbered.

Comment 2) Ristocetin per se is not physiologically relevant. Its study is of interest solely because it is generally believed that ristocetin simulates the effect of shear stress on von Willebrand factor.Instead, the authors state “Ristocetin is an activator of glycoprotein (GP) lb/IX/V to interact with von Willebrand factor,...”. (Lines 20-21) and “Ristocetin is known as an activator of GP lb/IX/V to interact with vWF…..,”. (Line 81) These statements are not supported by the cited reference (#15) and are not consistent with the prevailing concept that ristocetin is an activator of von Willebrand factor. In ristocetin cofactor activity assay, a laboratory test to measure the activity of von Willebrand factor in plasma, ristocetin is added to formalin fixed platelets suspected in test plasma to induce platelet aggregation. In this process, GPIb/IX/V is cannot be activated since the platelets are formalin fixed. If the authors are aware of any credible literature that supports their claim, it should be cited.

Our response;

We must apologyze the incorrect description about ristosetin refered from the Ref. 15 in the original manuscript, which was renumbered to 18 in the revised manuscript. As pointed out, ristocetin is an activator of not GPIb/IX/V but vWF binding to the GPIb/IX/V with shear stress-dependent interaction, referred  in the article.We rewrote the sentences in Abstract and Introduction in the revised manuscript as follows:

*Line 20-21;

"Ristocetin is an activator of von Willebrand factor to interact with glycoprotein (GP) Ib/IX/V."

*Line 79;

 " Ristocetin is known as an activator of vWF that interacts with GPIb/IX/V, like shear stress " 

Comment 3) The entire manuscript should be proof-read by someone with fluency in English. As an example, "Ristocetin is an activator of glycoprotein (GP)lb/IX/V to interact with von Willebrand factor, whichgenerates thromboxane A2....." (Lines 20-21)

Our response:

 The manuscript underwent English editing by Editage. The certification for English editing was attached.

Reviewer 2 Report

In this study, the author investigated the activation pathway of platelets by ristocetin and CXCL12. The study focused on understanding the specific pathway triggered by these activators and explored their inhibitory effects. However, it raises questions about the selection of only the RAC and Rho kinase pathways for investigation.

It is worth considering whether a proteomics study analyzing on the clots in experimental groups could provide additional insights and a clearer understanding of the downstream pathways and proteins involved in platelet activation induced by these activators. Such an approach could potentially offer valuable information and suggestions.

Additionally, the author's inclusion of a comprehensive gel of western blots for all experimental groups, even with a Coomassie Blue stain, could facilitate a comparison of protein profiles. This comparative analysis has the potential to provide highly useful information regarding the effects of these activators and their impact on platelet activation.

Author Response

Dear Reviewer #2,

Thank you for your fruitful comments. We revised the manuscript according to your comments as follows;

  • In this study, the author investigated the activation pathway of platelets by ristocetin and CXCL12. The study focused on understanding the specific pathway triggered by these activators and explored their inhibitory effects. However, it raises questions about the selection of only the RAC and Rho kinase pathways for investigation.

Our comment:

Thank you for your point. We investigated Rac and Rho based on our previous findings about the effects of ristocetin and combination of CXCL12 and collagen on the platelet activation as described in the original manuscript. 

  • It is worth considering whether a proteomics study analyzing on the clots in experimental groups could provide additional insights and a clearer understanding of the downstream pathways and proteins involved in platelet activation induced by these activators. Such an approach could potentially offer valuable information and suggestions.

Our comment:

We take your point that it is worth considering whether a proteomics study analyzing on the clots in experimental groups could provide additional insights and a clearer understanding of the downstream pathways as well as proteins involved in the platelet activation shown here. We added the description about the point in Discussion in the revised manuscript as follows:

*Line 316-318:

" It is worth considering whether a proteomic study analyzing clots in experimental groups could provide additional insights and a clearer understanding of the downstream pathways and proteins involved in platelet activation"

  • Additionally, the author's inclusion of a comprehensive gel of western blots for all experimental groups, even with a Coomassie Blue stain, could facilitate a comparison of protein profiles. This comparative analysis has the potential to provide highly useful information regarding the effects of these activators and their impact on platelet activation.

Our comment:

As pointed out, the inclusion of a comprehensive gel of western blots for all experimental groups could facilitate a comparison of protein profiles. To our regret, however, we are not able to provide them at this time, because the original membranes have not been preserved.
